# Elevation-dependent intensification of fire danger in the western United States

Mohammad Reza Alizadeh [1,2,3], John T. Abatzoglou [4], Jan Adamowski[2], Arash Modaresi Rad[5], Amir AghaKouchak [6,7], Francesco S. R. Pausata [3] & Mojtaba Sadegh [5] ✉

Studies have identified elevation-dependent warming trends, but investigations of such trends in fire danger are absent in the literature. Here, we demonstrate that while there have been widespread increases in fire danger across the mountainous western US from 1979 to 2020, trends were most acute at high-elevation regions above 3000 m. The greatest increase in the number of days conducive to large fires occurred at 2500–3000 m, adding 63 critical fire danger days between 1979 and 2020. This includes 22 critical fire danger days occurring outside the warm season (May–September). Furthermore, our findings indicate increased elevational synchronization of fire danger in western US mountains, which can facilitate increased geographic opportunities for ignitions and fire spread that further complicate fire management operations. We hypothesize that several physical mechanisms underpinned the observed trends, including elevationally disparate impacts of earlier snowmelt, intensified land-atmosphere feedbacks, irrigation, and aerosols, in addition to widespread warming/drying.

Mountains provide a variety of ecosystem services, including supplying about 50% of freshwater globally and an even higher fraction in mountainous arid regions (e.g., 70% of runoff in the western US)[1,2]. Orographic temperature and precipitation gradients in montane areas facilitate stratified vegetation belts[3], which promote local hotspots of biodiversity with a high degree of complexity that are particularly vulnerable to ecosystem changes in response to chronic (e.g., warming) and/or acute (e.g., wildfire) stressors[4]. Even small perturbations in these mountainous areas have large repercussions for hydrological and ecological processes, with cascading effects on downstream human-environmental systems[3,5–7].

A growing body of literature points to elevation-dependent trends in meteorological and land surface characteristics in montane regions of the world in response to the underlying warming signal[6–8]. In particular, more rapid warming of surface air temperature has been documented at higher elevations compared to that of lower elevations in many regions globally[7,9]. Similarly, trends in land surface temperature[10], reference evapotranspiration[11], snow cover and snow water equivalent[12,13], and vegetation greening[14,15] have also been documented to vary across elevational gradients. Furthermore, elevation-dependent precipitation trends are observed in some regions, although the sign of such changes and associated mechanisms vary across studies[2,16].

Changes in the energy and water balance alter the fire danger level[17], which is defined as the potential for a fire to ignite, spread and require suppression action. The literature has shown widespread increases in fire danger in many regions globally[18–21], but how fire danger trends change across the elevational gradient has not been studied. Recent literature shows that atmospheric warming weakened the high-elevation flammability barrier and enabled the upslope

[1]Department of Civil and Environmental Engineering, Massachusetts Institute of Technology, Cambridge, MA, USA. [2]Department of Bioresource Engineering, McGill University, Montreal, QC, Canada. [3]Department of Earth and Atmospheric Science, University of Quebec in Montreal, Montreal, QC, Canada. [4]Management of Complex Systems Department, University of California, Merced, Merced, CA, USA. [5]Department of Civil Engineering, Boise State University, Boise, ID, USA. [6]Department of Civil and Environmental Engineering, University of California, Irvine, CA, USA. [7]Department of Earth System Sciences, University of California, Irvine, CA, USA. ✉e-mail: mojtabasadegh@boisestate.edu

advance of fires[22], and facilitated high-elevation fires that are unprecedented in modern history[23]. Here we use the fire danger representation in the US National Fire Danger Rating System (NFDRS)[24] to investigate trends in fire danger across elevations. We note that NFDRS fire danger indices were empirically derived based on mathematical models of fire behavior, and they do not fully capture the energy and water balance related to fuels and fire[24]. Previous studies have linked these indices with regional burned area[25] and the growth of individual fires[26], and they are operationally used by fire management agencies across the US[27].

We evaluated elevation-dependent trends in fire danger indices between 1979 and 2020 across 15 level III Omernik ecoregions of the western US[28] that are mountainous. We augmented this list with a variety of meteorological variables that are commonly used in fire studies[29,30] and posed two questions: (1) Are there elevation-dependent trends in fire danger indices across montane regions of the western US? (2) If so, do they culminate in the elevational synchronization of fire danger? We answered these questions using the gridMET dataset of daily meteorological and NFDRS variables (~4 km grid)[31], Omernik level III ecoregions map from the US Environmental Protection Agency[28], and the National Elevation Dataset (10 m resolution) from the US Geological Survey. We present the results of the energy release component (ERC, Fuel Model G in NFDRS 77)[24] in the main paper and all other NFDRS indices (burning index [BI], 100-h and 1000-h dead fuel moisture—FM100 and FM1000, respectively) and meteorological variables (vapor pressure deficit [VPD], temperature, relative and specific humidity, and reference evapotranspiration) in the Supplementary Information. ERC indicates the available energy per unit area at the flame front, measuring the dryness of dead and live fuels. ERC is less directly influenced by temperature than other fire danger indices. It is, however, strongly influenced by relative humidity—which is in turn related to temperature—and precipitation. Here, we select a fuel-agnostic measure across western landscapes through a single fuel model (model G) not to conflate heterogeneity in vegetation distribution with heterogeneity in climate trends. We, however, show that general conclusions hold for other fire danger indices that are not dependent on the fuel model.

## Results

### Elevation-dependent trends in warm-season ERC

All fire danger indices, as well as meteorological variables, showed marked drying/warming trends over the period of 1979–2020 in all 15 mountainous ecoregions of the western US and across all elevation bands (Fig. 1, Supplementary Figs. S1–S8). Temporal trends in warm-season-average (hereafter warm-season; May–September) fire danger indices were computed over 500 m elevation bands in each ecoregion using least squares linear regression (e.g., Fig. 1a), and linear slope of trends was calculated across elevation bands (e.g., Fig. 1b). The former indicates temporal trends in fire danger indices in each band, whereas the latter points out whether or not trends are magnified at higher elevations compared to the lower land.

Warm-season ERC trends were most pronounced at higher elevations and least pronounced at lower elevations (Fig. 1a). Median ERC among all ecoregions increased by nearly 15 units during 1979–2020 in the highest elevation band (>3000 m). By contrast, median ERC across all ecoregions increased by only ~6 units—smallest across all elevation bands—during 1979–2020 in the lowest elevation band (0–500 m) (Fig. 1a). Among individual ecoregions and elevation bands, the largest increase in ERC (17 units) from 1979 to 2020 was observed at >3000 m in Central Basin and Range, whereas the smallest increase in ERC (1 unit) was observed in the 0–500 m elevation band of the maritime affected North Cascades (Fig. 1a).

Positive elevation-dependent ERC trends (i.e., larger increases in ERC with elevation gain) were found in 13 of the 15 studied ecoregions (Fig. 1b). This trend is even more pronounced for dead fuel moisture (FM100 and FM1000), with all ecoregions showing more marked drying trends in higher elevations (Supplementary Figs. S3, S4). Positive elevational ERC slopes range between 1 and 3 units/km in the 42 years of study (Fig. 1b). For Central Basin and Range, for example, the highest elevations (>3000 m) had an additional 6.4 units of increase in ERC compared to lower elevations (1000–1500 m) from 1979 to 2020. Accelerated increases in fire danger at higher elevations imply synchronization of fire danger across elevations, posing marked fire management challenges[22,32–34].

Warm-season average ERC in the most recent decade (2011–2020) was larger than that of the earliest decade (1981–1990) in all ecoregions

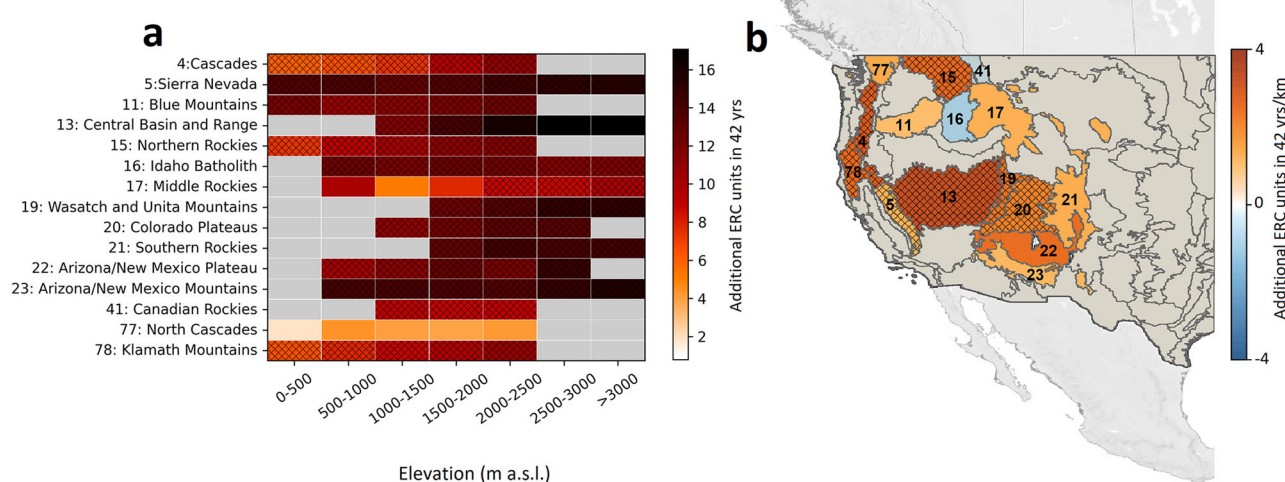

**Fig. 1 | Elevation-dependent trends in fire danger across montane ecoregions in the western US. a** Temporal trends in warm-season (May–September) average energy release component (ERC) from 1979 to 2020 in each elevation band in each mountainous ecoregion of the western US. Elevation bands with less than 250 km² of land are removed from the analysis and are shown with gray shading. **b** Slope of ERC temporal trends across elevation bands, where positive values indicate a larger

intensification of ERC at higher elevations. (© OpenStreetMap contributors 2017. Distributed under the Open Data Commons Open Database License (ODbL) v1.0.)[57]. Gray shading shows non-mountainous ecoregions that are not studied here. Hatched areas indicate statistically significant trends at the 95% confidence level. "m a.s.l." stands for meter above sea level, and "yrs/km" stand for years per kilometer of elevation.

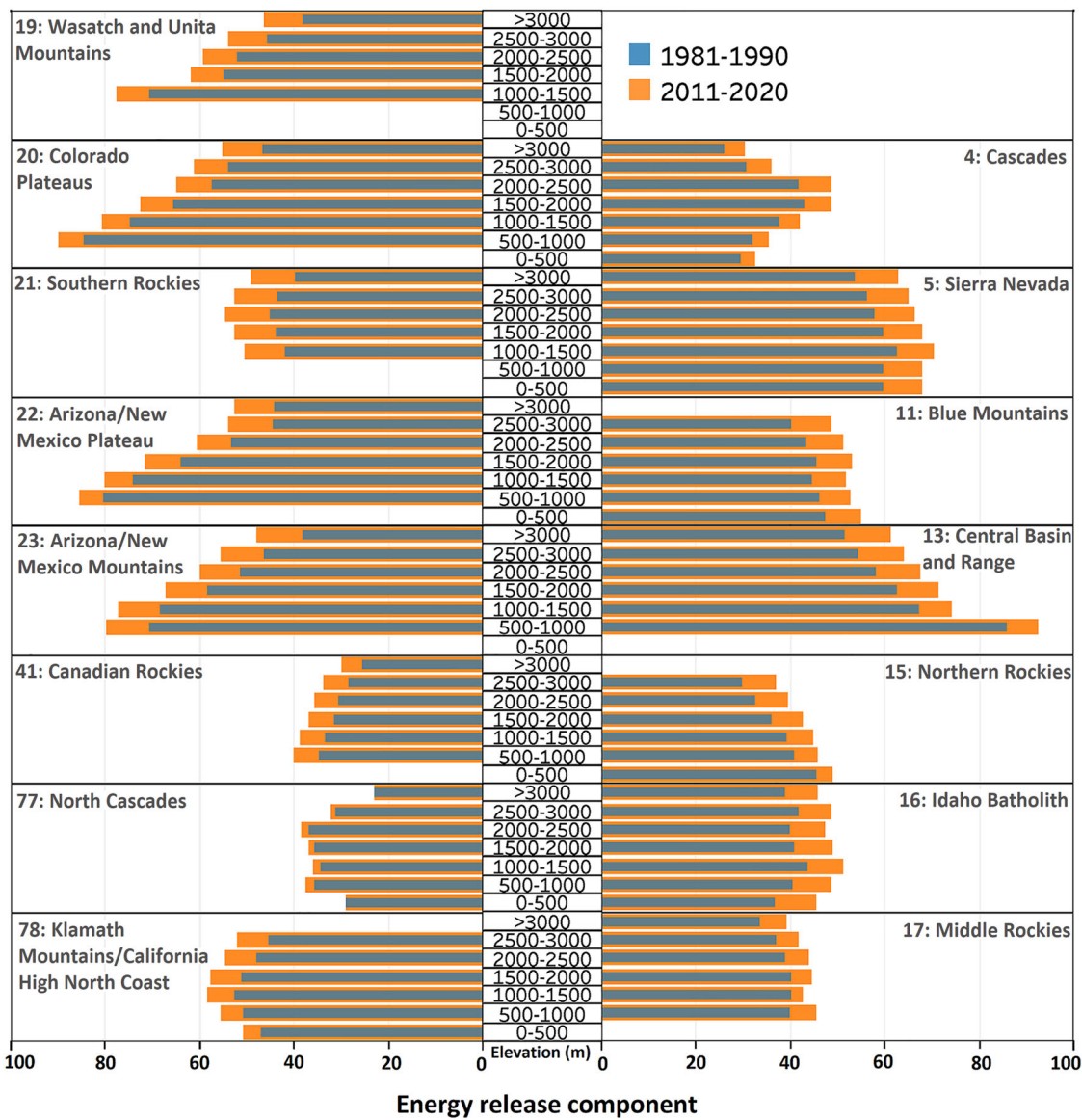

**Fig. 2 | Decadal average warm-season energy release component (ERC) in each elevation band in each ecoregion.** Results for 1981–1990 and 2011–2020 are shown in blue and orange colors, respectively. "m" stands for meter.

and across all elevation bands (Fig. 2). Similar drying/warming behavior was observed when viewed through the lens of other fire danger indices and meteorological variables (Supplementary Figs. S9–S17). The largest median relative increase in warm-season ERC (19%) during 2011–2020 vs. 1981–1990 across all ecoregions was observed in the highest elevations (>3000 m), and the smallest relative increase (10%) was observed in the lowest elevations (0–500 m). Furthermore, a range of different patterns are observed in warm-season ERC values across different elevation bands within and between ecoregions (Fig. 2). Some ecoregions (e.g., Sierra Nevada) were associated with a comparable range of warm-season ERC values across elevation bands, whereas others (e.g., Central Basin and Range) showed widely different values across the elevational gradient (Fig. 2). In general, the elevational gradient of warm-season ERC climatology is more pronounced in drier/warmer ecoregions (Fig. 2). Southern ecoregions (e.g., New Mexico Plateau/Mountains) were expectedly associated with a higher warm-season ERC range compared to northern ecoregions (e.g., Canadian Rockies), which follow latitudinal temperature gradients (Fig. 2).

Warm-season average temperature generally decreases monotonically with elevation gain, but elevational relationships of fire

danger indices are non-monotonic (Fig. 3). Fire danger indices depend on the energy balance driven by moisture availability, evaporative demand, and temperature (as well as wind for BI) and are hence not merely a simple function of temperature. Responses of ERC, BI, and VPD, as well as FM100, FM1000 range from (1) monotonically decreasing (increasing for FM100 and FM1000) with elevation (Fig. 3a; Arizona/New Mexico Mountains) to (2) increasing (decreasing for FM100 and FM1000) in response to elevation gain (Fig. 3b; Cascades), and (3) increasing (decreasing for FM100 and FM1000) to a certain elevation band and a reversed trend afterward (Fig. 3c; Sierra Nevada). We hypothesize that lower temperature and higher moisture availability in higher elevations in Arizona/New Mexico mountains lead to the monotonic decline in fire danger with elevation (Fig. 3a). By contrast, the lower elevation western slope of the Cascades is impacted by maritime air mass leading to reduced fire danger indices compared to higher elevations (Fig. 3b). In Sierra Nevada, lower elevations are adjacent to California's Central Valley that is heavily irrigated and promotes elevated humidity in the boundary layer that moderates ERC[35], whereas lower humidity in mid-elevations (1000–2000 m) on the western slope of the region fosters the most intense fire danger

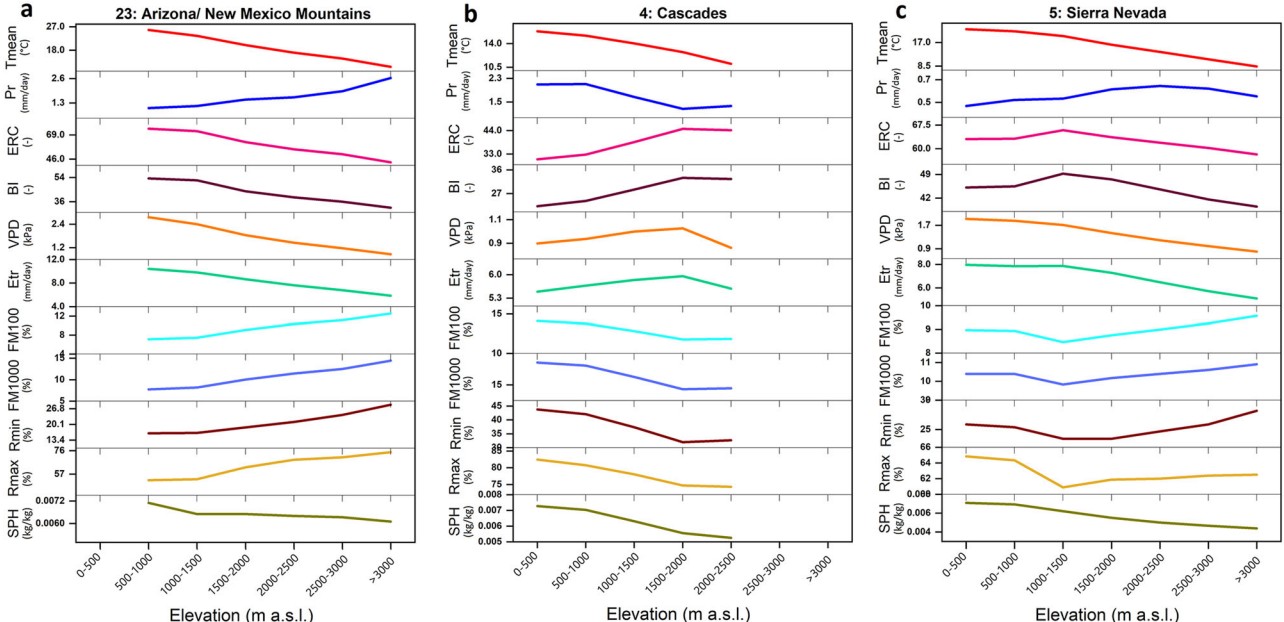

**Fig. 3 | Elevational changes in the climatology of meteorological variables and fire danger indices.** Average warm-season values of mean daily temperature (Tmean), precipitation (Pr), energy release component (ERC), burning index (BI), vapor pressure deficit (VPD), daily reference evapotranspiration (based on alfalfa; Etr), 100-h and 1000-h dead fuel moisture (FM100 and FM1000, respectively), minimum and maximum daily relative humidity (Rmin and Rmax, respectively), and specific humidity (SPH) from 1979 to 2020 in each elevation band are presented for **a** Arizona/New Mexico Mountains, **b** Cascades, and **c** Sierra Nevada. "m a.s.l." stands for meter above sea level, "mm" stands for millimeter, "kg" stands for kilograms, and "kPa" stands for kilopascal.

indices while decreased temperature and increased moisture (e.g., due to orographic increase in precipitation and snow cover) promote reduced fire danger indices at higher elevations (>2000 m). Other ecoregions are shown in Supplementary Figs. S18–S29.

**Elevation-dependent increase in critical fire danger days**

We now turn our attention to critical fire danger days, which are associated with high fire activity and potential for fire growth. We considered a threshold of ERC = 60 as the tipping point for increased fire activity across all studied ecoregions, following Brown et al.[36] that showed a majority of large forest fires (>400 ha) in the western US started on days with ERC ≥ 60. Our analysis of fire records in the Fire Occurrence Database[37] confirmed this reporting and showed that 77 and 83% of large (>400 ha) and very large (4000 ha) fires from 1992 to 2020 in the studied ecoregions were associated with ERC ≥ 60 on their discovery date. These statistics also hold for the entire western US. The constant threshold of ERC = 60 is selected here to warrant consistency and inter-comparability across ecoregions.

We found an increase in critical fire danger days during 1979–2020 in all elevation bands and all ecoregions (Fig. 4a). The highest median increase in critical fire danger days in all ecoregions during the 42 years of this study occurred between 2500 and 3000 m with an overall increase of 63 days, of which 22 days occurred outside of the warm season (Supplementary Tables S1–S3). By contrast, the lowest median increase in critical fire danger days across all ecoregions occurred in the 0–500 m elevation band, adding >22 extra fire danger days in 42 years (Supplementary Tables S1–S3; Fig. 4a). Higher elevations in 10 of the 15 studied montane ecoregions were associated with a larger rate of increase in critical fire danger days compared to lower elevations (Fig. 4b). The highest slope of trend in critical fire danger days as a function of elevation was observed in the Central Basin and Range, indicating an additional >28 fire danger days in 42 years per 1 km of elevation gain (Fig. 4b).

Trends in critical fire danger days viewed through the lens of other fire danger indices follow a similar pattern as that of ERC (Supplementary Figs. S30–S32).

The largest median rate of the relative increase in annual critical fire danger days (119%), based on ERC, across all ecoregions during 2011–2020 as compared to 1981–1990 was observed at >3000 m, and the lowest relative increasing rates were observed at <1500 m ranging between 38 and 43% (Fig. 5). Results based on other fire danger indices follow a similar pattern (Supplementary Figs. S33–S35). Furthermore, critical fire danger days synchronized across all elevation bands in the most recent decade in many ecoregions, such as Central Basin and Range and Sierra Nevada, indicating lessened topographical fire danger relief in a warming climate (Fig. 5). However, the decrease of critical fire danger days with elevation gain, due to the lower baseline ERC at higher elevations (Fig. 2), is noted in multiple ecoregions, such as Wasatch and Unita Mountains (Fig. 5). Finally, the number of critical fire danger days across ecoregions follow the latitudinal temperature gradient with the highest occurring in the southern region and the lowest in the northern region (Fig. 5). This is expected given baseline ERC values are higher in the southern ecoregions and our adopted fire danger threshold (ERC = 60) is constant across all studied ecoregions.

Finally, to augment this analysis, we also used the 75th and the 95th percentiles of daily ERC records from 1979 to 2020 in each grid pooled over the entire calendar year as the threshold for high and extreme fire danger conditions (Supplementary Fig. S36). We then estimated the number of high and extreme fire danger days in each grid and averaged them in each elevation band, and replicated our trend analysis. Results (Supplementary Figs. S37–S40) confirmed the findings of the constant threshold-based analysis (ERC = 60), although nuanced differences exist between the two approaches—especially between those of the constant and the 95th percentile-based thresholds. Furthermore, we conducted this percentile-based threshold analysis for other variables (Supplementary Figs. S41–S52) with similar conclusions.

## Discussion

Here we documented larger fire danger trends at high elevations compared to low elevations across mountainous ecoregions in the western US over the past four decades. Our results pointed to the synchronization of fire danger across elevations in many ecoregions

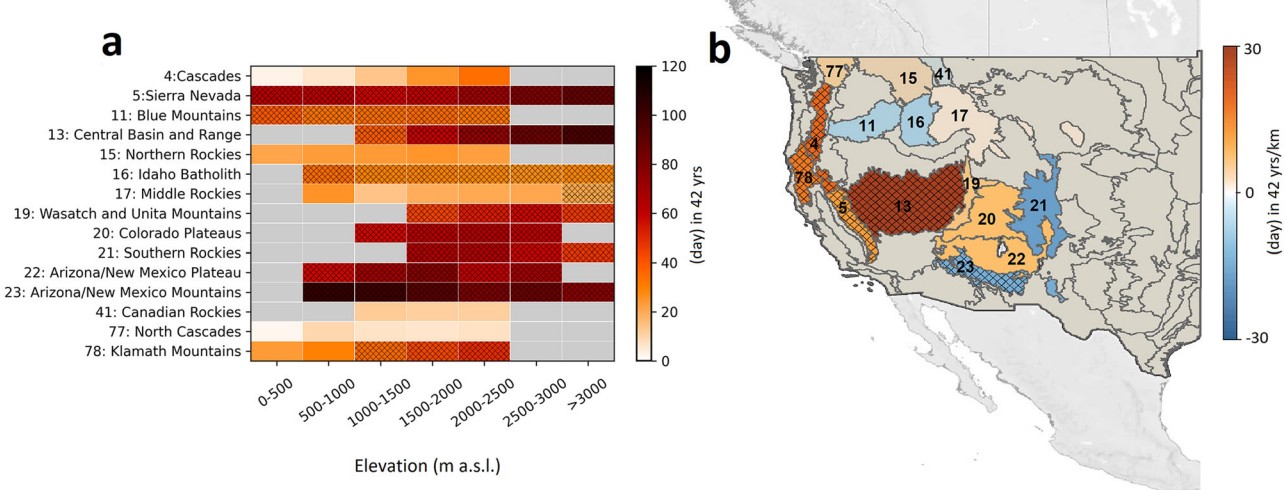

**Fig. 4 | Elevation-dependent increase in critical fire danger days. a** Temporal trends in critical fire danger days from 1979 to 2020 in each elevation band and ecoregion. **b** Slope of temporal trends in critical fire danger days across elevation bands. (© OpenStreetMap contributors 2017. Distributed under the Open Data Commons Open Database License (ODbL) v1.0.)[57]. Hatched areas indicate statistically significant trends at the 95% confidence level. "m a.s.l." stands for meter above sea level, and "yrs/km" stand for years per kilometer of elevation.

and indicated the reduction and disappearance of topographical fire danger relief in a warming climate. Elevation-dependent fire danger intensification implies that higher elevations that were historically wet enough to buffer fire ignition and slow/hinder fire propagation have become conducive to large fire activity in recent decades[22]. This trend is expected to intensify further, given the projected warming and drying trends in the western US[38]. Elevational synchronization of fire danger along with the documented spatial synchronization of fire danger across the western US forests[19] implies further strains on the limited fire suppression and management resources. We also documented concerning trends in critical fire danger days, especially at higher elevations. Our results showed that a marked portion of this increase in critical fire danger days occurred outside of the warm season, particularly in the southern ecoregions with a higher ERC climatology baseline.

We recognize that ground observations of meteorological variables are rare at high elevations, which might induce uncertainty in the reported trends based on a gridded product (gridMET[31]). Our analysis of fire danger trends across the elevation gradient using ground observations, however, confirmed the reported findings (Supplementary Fig. S53), although for a limited number of ecoregions (five) and stations (a total of 79) constrained by data availability. We also note that elevation-dependent warming has been widely demonstrated across the globe, including in the mountains of the western US (Pepin et al.[6,7] and references therein); and this trend on the backbone of widespread drying in the fire season[39] is expected to induce elevation-dependent fire danger intensification. Furthermore, Alizadeh et al.[22] showed that normalized burned area in the high-elevation forests of the western US increased at a higher rate than its low-elevation counterpart from 1984 to 2019, pointing to a weakened flammability barrier in the high elevations, providing secondary evidence for the herein reported trends.

We hypothesize that several mechanisms contribute to the elevation-dependent trends in fire danger. Earlier snowmelt and shrinking of snow cover decrease albedo at high elevations that historically stored large snow packs. This contributes to surface warming as a result of increased absorption of incoming solar energy[7]. While earlier snowmelt may not have directly been the main contributor to the largest intensification rates of fire danger at higher elevations, especially in the warm season, the indirect impact of earlier snowmelt can contribute to soil desiccation and land-atmosphere feedbacks strengthening that intensify fire danger[40,41]. Land-atmosphere feedbacks have always been a driving factor in water-limited low-elevation regions but have historically been less ubiquitous in energy-limited, moist highlands that have observed the largest soil moisture decline rates in recent decades in response to warming[38]. The warming and drying cycle, due to land-atmosphere feedback, is further intensified by inhibiting cloud formation and its associated energy balance effects, as well as increasing the boundary layer depth that traps heat in the atmosphere[40]. Similarly, warming and drying lead to higher cloud base heights, reducing the total precipitation that reaches the ground[42].

Aerosols also play a role in the observed trends, as valleys of the western US trap fire smoke, dust, and anthropogenic particles and change long- and short-wave radiative balance[7,43]. Higher concentrations of aerosols in the valleys buffer the direct impact of the incoming shortwave radiation on the surface weather[44]. This enables the surface air temperature to be cooler than its potential[45]. The aerosol impact is lower at high elevations[46]. Another significant contributor to the smaller intensification of fire drivers in the lowest elevation bands in some ecoregions is the impact of agricultural irrigation on regulating valley temperatures[35].

Elevation-dependent intensification of fire danger has important implications for future ecological and hydrological characteristics of montane ecosystems[34]. High-elevation mesic forests are associated with long return interval (several decades to millennia), high-intensity, stand-replacing fires, and their frequent occurrence may alter the population, community, composition, and structure of these forests[47–49]. Fire impacts compounded by a warming climate also threaten high-elevation plant species by facilitating pathways for low-elevation species, including invasive annual grasses, to move to upper ground[50]. Increasing high-elevation fire activity also has significant implications for: (1) water availability through removing vegetation cover and impacting snow accumulation and melt[34,50], (2) water quality, through introducing various pollutants and facilitating a magnified increase of stream temperature[7,51], and (3) landscape morphology, through enhanced erosion rates and stream incision[52]. In-depth understanding of elevation-dependent trends in fire danger is specifically important as fire suppression efforts are least effective in high-elevation, mesic forests, which when burned can significantly impact the vulnerable highlands' flora and fauna, and can have adverse effects that cascade to lower elevations that depend on high-elevation emanated ecosystem services[53].

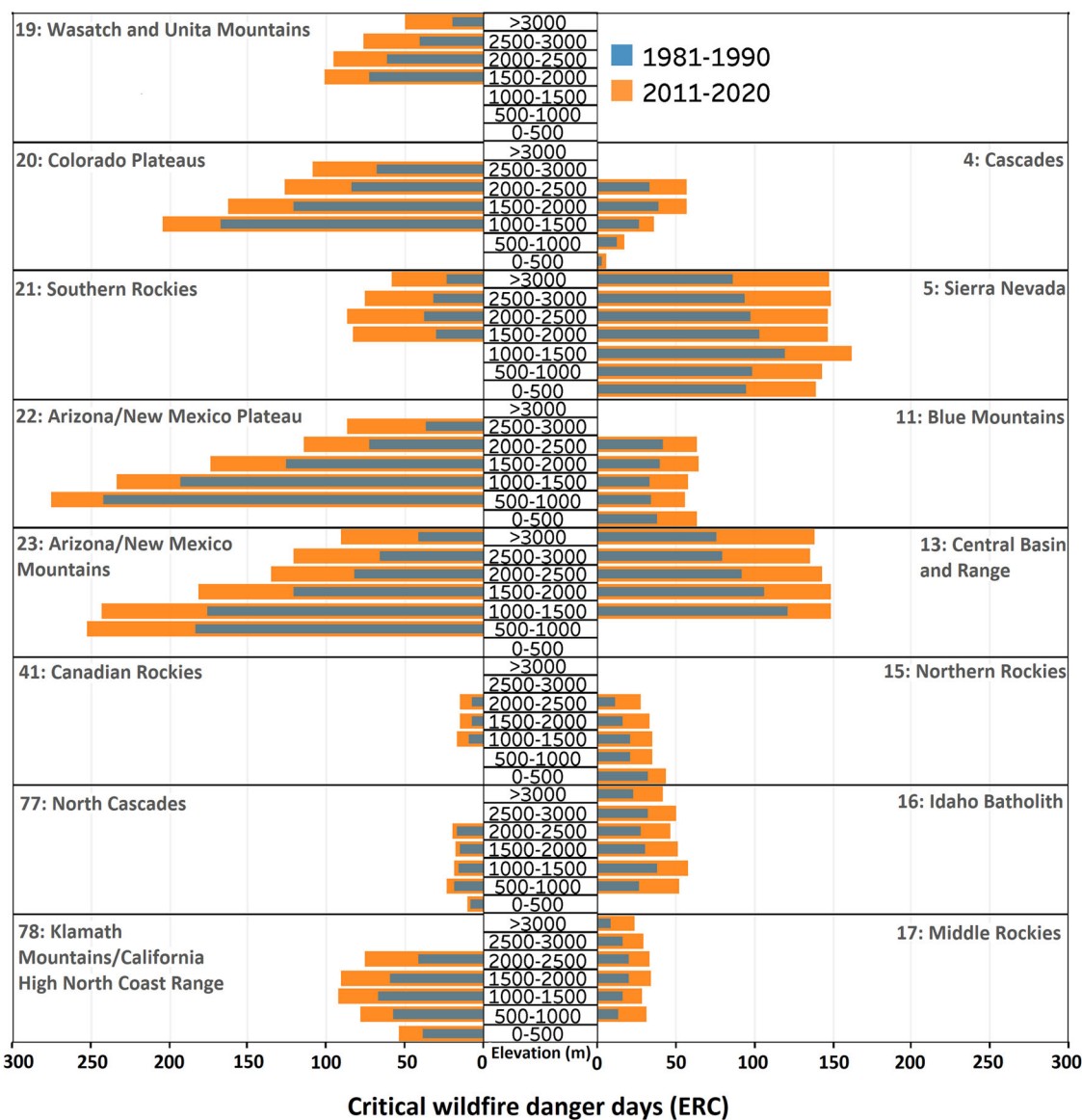

**Fig. 5 | Annual critical fire danger days.** Decadal average critical fire danger days per year based on energy release component (ERC) ≥ 60, during 1981–1990 (blue) and 2011–2020 (orange). "m" stands for meter.

## Methods

We calculated the warm-season average of meteorological and US National Fire Danger Rating System indices, herein referred to as fire danger indices, using daily values in each grid from the gridMET[31] dataset (~4 km resolution), and then averaged them for each 500 m elevation band in 15 mountainous ecoregions of the western US. We selected Omernik level 3 ecoregions[28] that encompass mountain ranges of the western US. We then used these variables for (1) estimating least squares linear trends in warm-season averages and (2) quantifying the number of critical fire danger days. We considered fire drivers in each ecoregion separately since each ecoregion includes rather similar ecoclimatic characteristics. We divided each ecoregion to land encapsulated in 500 m elevation bands (e.g., band 1: 0–500 m above sea level, a.s.l., band 2: 500–1000 m a.s.l., ..., band 7: >3000 m a.s.l.) to investigate elevation-dependent trends in various biophysical and atmospheric variables. We removed bands with <250 km² of land (less than 16 grid cells in gridMET) from the analysis to ensure robust results. Supplementary Table S4 lists the surface area encapsulated in each elevation band in each ecoregion.

We used daily average temperature, precipitation, vapor pressure deficit (VPD), energy release component (ERC), burning index (BI), 100-h dead fuel moisture (FM100, representing small diameter fuel), 1000-h dead fuel moisture (FM1000, representing large diameter fuel), minimum and maximum daily relative humidity (Rmin and Rmax, respectively), and specific humidity (SPH). We used the US National Fire Danger Rating System 77[24] for the calculation of fire danger indices. Furthermore, we employed the Fuel Model G (dense conifer stands) for ERC and BI calculations over the entire western US landscapes as a fuel-agnostic measure for documenting climate-driven fire danger trends, not to conflate heterogeneity in vegetation distribution with heterogeneity in climate trends. We defined the warm season as May–September since this period is associated with enhanced fire activity in the western US. For warm-season trends, we used the May–September average of each variable. We present the results based on ERC in the main paper and other variables in the Supplementary Information.

For critical fire danger days, we counted the number of days in which the daily variable exceeded its defined threshold. This threshold is selected from the literature and is associated with increased fire

activity and growth potential[36,54]. Threshold values were selected as: ERC = 60; FM100 = 8%, FM1000 = 10%, and VPD = 2 kPa[54,55]. We augmented this analysis with a local percentile-based threshold for high and extreme fire danger days, in which the 75th and 95th (25th and 5th for fuel moisture) percentiles of long-term daily time series of various variables in each grid pooled over the entire calendar year were selected as the threshold, and the number of high and extreme fire danger days in each grid in each year was estimated accordingly. Grid estimates were then averaged over the entire elevation band, which were subsequently used for trend analyses. In all analyses, the slope of linear least squares regression was presented. The underlying warm-season ERC data justified the use of linear trends (Supplementary Fig. S54), but we acknowledge that not all variables necessarily satisfy the assumptions of a linear regression analysis. We provided warm-season averages of all variables in the Supplementary Data to enable more in-depth analyses. Finally, the two-sided $t$-statistic was used to test the null hypothesis that the slope coefficient of linear regression is equal to zero. Upon rejection of the null hypothesis ($p$-value ≤ 0.05), we accept the alternative hypothesis that the trend is significant.

## Data availability
The referenced climate and fire danger data can be obtained from the gridMET dataset available at: https://www.climatologylab.org/gridmet.html. The referenced Omernik ecoregion boundaries are available at: https://www.epa.gov/eco-research/level-iii-and-iv-ecoregions-continental-united-states. The referenced elevation data are obtained from the National Elevation Dataset, which is available at: https://www.usgs.gov/the-national-map-data-delivery. The processed elevation-dependent warm-season fire danger indices are available in the Supplementary Data 1 file.

## Code availability
Source codes are available at: Alizadeh (2022)[56]. alizadeh-mr/Wildfire-danger-indices: Initial Release (v1.0.0). Zenodo, (https://doi.org/10.5281/zenodo.7424926)

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

## Acknowledgements

We acknowledge support from the US Joint Fire Science Program grant number L21AC10247 (M.S. and J.T.A.) and the Fonds de recherche du Québec – Nature et Technologies (FRQNT) number (B3X-321272) (M.R.A.). We appreciate the Google Earth Engine and Google Collaboratory platforms for the computational resources.

## Author contributions

M.S., M.R.A., and J.T.A. conceived the study and wrote the first draft of the manuscript. M.R.A. conducted all the analyses. J.A., A.M.R., A.A., and F.S.R.P. contributed to the analysis and final paper.

## Competing interests

The authors declare no competing interests.
