## [Peer Review File · Nature Communications]

Elevation-dependent intensification of fire danger in the western United StatesEditorial Note: Parts of this Peer Review File have been redacted as indicated to remove third-party material where no permission to publish could be obtained.

REVIEWER COMMENTS

Reviewer #1 (Remarks to the Author):

This paper seeks to define the presence and, if present, magnitude of trends in fire danger indices in the western U.S. by elevation. It presents evidence that this may be the case, but that evidence is based on poorly described methods and has caveats altogether undescribed and thus dismissed.

In particular, the authors rely on a gridded climatic dataset (gridMet) that is itself a blend of other inputs that are required to calculate the various fire danger indices. The gridMet documentation online and in Abatzoglou 2013 indicate that the gridded datasets likely used to produce the derived fire danger indices are a combination of NLDAS-2 and PRISM, so rely in part on base data that estimate the temperature and precipitation (and other derived or, in the case of NLDAS, supposedly physically consistent) differences at high elevations from interpolation rather than observation; dramatically fewer high elevation stations exist, and so validation is comparatively limited. The authors make no effort in the current paper to diagnose whether the elevation dependence inferred to be “real” could be instead an artifact of interpolation schemes, changes in data availability, or the “blending” and downscaling processes acting on differentially assimilated data through time. I am sympathetic that they are using the best there is for these purposes, but also think reanalysis, rescaling, and blending must be shown (or logically defended) to not produce the results as artifacts. I also get it that as a heavily derived variable, ERC can’t really be verified, only computed. But the fact remains the higher elevation responses are not validated here – only calculated and trends described as important.

I think the paper should include some methodological upgrades to: (1) more clearly state why the authors believe the trends computed have no artifacts (temporally inconsistent data assimilation / observational / timing / interpolation) contributing to the trends; (2) state how the trends were calculated (and why linear methods are appropriate – a number of these variables lend themselves to non-standard distributions); and (3) describe and then discuss the limitations of the fuel model(s) chosen for ERC.

Moreover, the paper focuses on ERC almost totally at the expense of other drivers of “fire danger”, and as such an important caveat of the work is that it addresses primarily the fire intensity aspects of fire danger, which is much different than assessing the other aspects of fire danger people typically think of – fire spread, fire weather, etc. – and to which fire managers respond. Stating this in the text would be valuable.

Line by line comments:

Line 36: In-text citation here, but Vancouver style citations elsewhere?

Lines 43-44: “. In particular, more rapid warming of surface air temperature has been documented at higher elevations compared to that of lower elevations in many regions globally^{5,9}” The way this sentence is written implies analysis of observations, but citation 9 is a modeling paper only. I suggest expanding this into two supportable statements, one that it seems to be actually occurring based on data (5) and that the mechanisms for it largely point to snow feedbacks but manifest differently with different LSMs (9). Or similar.

Line 47: There is no single elevation gradient at play here, nor are these changes universally documented, so it should be elevation gradients (plural).

Line 50: NFDRS describes variables that are themselves related to water and energy balance, but a key limitation of NFDRS is that it relies on a few variables, correlated with fire danger as measured by people and parts of energy and water balances, but NFDRS does NOT include the full energy and water balance as they relate to fuels and fire. As such, this is imprecisely stated and needs to be refined to be true. One outcome of changes in energy and water balances is changes in fire danger, but they are not as directly related as this text makes it seem.

Line 63: Does...trends ◊ Do...trends

Line 64: Gridmet citation required here – is in methods too, but should be stated here.

Line 67, and in methods. ERC calculations depend on the fuel model chosen. Ecoregions should have different mean vegetation assumptions, and so fuel models could be presumed to vary. In addition, ERC weights larger fuel classes (100hr and 1000hr) more heavily than fine fuels that drive fire spread, and the fire danger aspect of ERC is primarily related to fire intensity (energy released at the flame front). ERC, if evaluated for a universal fuel class (I presume 7G or H, which are common?) across these regions would differentially bias trends of actual fire danger because absolute changes in mean or extreme conditions required to exceed historical thresholds would be quite different for fuel models such as 1 (grass) or 4 (chapparal). It is also true that trends in fuel structure over the period of the study could bias the results as well. It is possible that because the trends are compared for spatially aggregated units and fuels changes were ignored the presence or absence of a trend would remain even if the intercept were incorrect. The authors need to at minimum (1) describe in the methods how they handled the different fuel model possibilities and (2) state why (if it does not) that has no repercussions for results. If it does have implications for results, the authors need to correct the analysis, presumably by specifying different mean fuel models for the ecoregions.

Line 69-71: ERC has a 7 day memory as typically computed. Especially for coarser fuels (1000hr) this means

Line 154: alfaalfa \diamond alfalfa

Line 159-164: Except for the fact that BI includes both, the authors seem to ignore the fact that NFDRS has both ERC and SC – the spread component is part of what determines how likely fires are to grow quickly and burn large areas. ERC will be correlated with the initial potential for intensity to achieve values sufficient to essentially guarantee re=heating and combustion of other fuels in close proximity, but the rate of spread and eventual size of fire is dependent on factors (wind, especially) associated with fire behavior. Critical fire danger days are NOT merely ERC >60, in part because even if all high danger events ERC>60, not all ERC>60 result in high danger. By focusing on the strongest evident trend, the potential utility of the science is undermined.

Line 211, again 245: “herein revealed”....that language sounds more like marketing than science. I would steer clear of it, especially since the details may, or may not, support revelation.

Line 263: “We used geospatial analysis....” Is very, very general. Do you mean to say you took daily ecoregional averages by these elevation bands?

Line 266: “estimating linear trends in warm-season averages”. There are a lot of different ways to estimate linear trends, both magnitude and significance. The choice matters, though, because the nature of the variable responses is by no means guaranteed to be linear and the statistical significance (variability versus magnitude) of the trend is critical. Some specifics here would help make this more presentable.

Reviewer #2 (Remarks to the Author):

Widespread warming/drying is evident, but not consistently so, across western US landscapes over recent decades. For example, underlying/related trends (i.e., in surface temperature, precipitation, evapotranspiration, snow cover, greenness) have been found to vary across the elevational gradient. These changes should be reflected in elevation-related trends in fire danger. Geographic specificity with regard to increased/increasing fire danger is important to factor into fire and land management planning and response aimed at mitigating risk to highly valued resources and assets. The research presented here takes a deeper dive into demonstrated increases in fire danger, focusing on the past four decades of change across the elevation gradient in mountainous regions of the western US. Findings include a remarkably consistent signal of increased warm-season fire danger across all metrics analyzed, with greatest changes over the study period in the highest-elevation band (>3000 m). Estimated numbers of days conducive to large-fire activity

increased the most at 2500-3000 m (present in nine of the 15 focal ecoregions). In some ecoregions, including the Central Basin and Range, the estimated average annual number of critical fire-weather days is now essentially equal across the elevation gradient. The work is straightforward (using the gridMET dataset of daily meteorological and NFDRS variables, Level III EPA Ecoregions, and 10-m data from the National Elevation Dataset) and thoroughly yet succinctly reported, with ERC as the focal fire danger index but similar changes visible through the lens of each fire-danger metric (included as supplementary material). (The gridMET dataset has been used extensively to explore a variety of connections between wildfires and climate.) The findings, especially that relatively mesic higher-elevation zones in mountainous regions have become fire receptive over longer within-year periods in recent decades, have important ecological and fire-management implications.

My primary concerns/suggestions are as follows:

Data and methodology

Lines 164-165, 283: Brown et al. (2004) is cited as the source of the critical fire danger threshold for used for ERC. As the authors are aware, an ERC of 60 does not mean the same in terms of percentile fire danger across the west or even within an ecoregion. An ERC value of 60 may indicate 95th percentile conditions in one cell, whereas in another cell the 95th percentile ERC value may be 89 – but in both cases the 95th percentile value indicates a comparable level of aridity for that microclimate (e.g., Riley et al. 2013). Figure 2 in Freeborn et al. (2019) shows the wide range of median and maximum ERC(G) values calculated from 39 years of gridded data. This is why contemporary geospatial modeling applications and Fire Danger Operating Plans, for example, are not tied to an absolute index value like ERC = 60 for any of their thresholding, but instead use historical analyses of fire activity and weather to set ERC breakpoints and associated adjective ratings (e.g., “low, medium, high, extreme” fire danger). The authors explain that their “constant threshold, as opposed to a percentile-based approach, is selected herein to warrant consistency and inter-comparability across ecoregions.” But I would like to see a bit more discussion about the pros and cons of the specific ERC = 60 threshold use. Has there been any further investigation since the Brown et al. (2004) publication that supports widespread use of ERC \geq 60 as a critical fire danger level? For example, have there been efforts that this work could capitalize on to associate historical large fire activity more precisely with ERC thresholds better tailored to the analysis units herein? Or could the methods here be expanded to make refinements to the ERC threshold as warranted? Perhaps some sensitivity analysis could be performed? Since I expect the media and others will pick up on the reported finding that “the greatest increase in the number of days conducive to large fires occurred at 2,500-3,000 m, adding 63 critical fire days in 42 years”, that seems like a particularly relevant slice of the analysis to look a bit deeper into the relationship between historical fire activity and ERC \geq 60.

(Riley, Karin L.; Abatzoglou, John T.; Grenfell, Isaac C.; Klene, Anna E.; Heinsch, Faith Ann. 2013. The relationship of large fire occurrence with drought and fire danger indices in the western USA, 1984-2008: The role of temporal scale. *International Journal of Wildland Fire*. 22: 894-909.)

Lines 264-265: For those who may not be familiar with EPA ecoregions, how were the “15 mountainous ecoregions” selected? Is “mountainous” an EPA ecoregion attribute?

Line 275: Which NFDRS version and fuel model was used in calculations? Because ERC is the focus of the main analysis/discussion, it in particular merits further explanation as to what it represents and how it was calculated (e.g., following Freeborn et al. 2015, Jolly et al. 2019).

Suggested improvements

Figures S9-S18 are missing the ecoregion labels in each box.

Line 76: Statement refers to “all fire danger indices” but cites only Figs. S1-S8. ERC data are presented in Fig. 1, which should be referenced here as well.

Line 78: All figures (1, S1-S8) are presented with the same two panels (A&B); suggest referencing

Fig. 1A and 1B here as examples "(e.g., Fig. 1A)."

Lines 122-125, 194-196; Fig. 2, S9-S18. The placement of each ecoregion's plot in these figures appears based on (information-free) ecoregion number: from lowest-to-highest, top-to-bottom, starting on the right panel. Have you considered placement that aligns better with the N-S-E-W location (or just latitude) in the western US? For example, Arizona/New Mexico Mountains and other southern ecoregions toward the bottom of the figure and Canadian Rockies, North Cascades, et al. toward the top? There wouldn't be perfect correspondence, of course, (you could use the current logic for "ties") but it would help convey general geographic differences in ERC values (e.g., appearing higher in the south vs. north) and how decadal averages have changed in more of a geographic context.

Clarity and context

Consider including a map showing the different elevation bands from the NED for the ecoregions used in the analysis. Or at least a map and/or size estimate of the area reported to have had the greatest increase in the number of days conducive to large fires (i.e., 2,500-3,000 m within the nine ecoregions).

Line 191: Some results are indicated to reflect "disappearance of topographical relief in a warming climate." While there may be lessened "relief" from fire danger with elevation gain (cool double meaning), the elevational gradient persists. So perhaps reword to say "disappearance of topographical relief in a warming climate from a fire-danger standpoint" – or something to that effect. Referred to as "fire danger relief" in Line 209.

Line 208: Is "synchronization" the best term here? It makes it seem as though things are happening all in unison now. But it's more nuanced than that. Pretty strong dryness gradients still exist in most ecoregions.

References

References 52 and 53 citations of the same manuscript.

Lines 162-165: Brown et al. (2004) is cited to support the assertion that the ERC = 60 threshold is "used by fire managers for devising management strategies and planning." However, this appears based on a statement by Brown et al. (2004) that "conversations with fire specialists independent of this study indicate that ERC values of 40 and 60 might be useful thresholds that can be related to management strategies and planning." Historical fire-weather analyses have advanced considerably over the last ~20 years. In keeping with previous comments, I would like to see a more contemporary citation that supports the assertion that an ERC = 60 threshold is used in management or planning applications today. Looking at the relevant fire danger plots here, https://www.predictiveservices.nifc.gov/fuels_fire-danger/national_fire_danger.html, could be informative, but they appear to be mostly based on NFDRS 2016/NFDRSv4 and Fuel Model Y, which goes back to the need to understand how ERC was calculated for this analysis.

Point-by-Point Response to Review Comments

The authors would like to thank the two anonymous reviewers that provided constructive comments and suggestions. In the following, the issues raised are addressed point-by-point in the order they are asked. Reviewer's comments are shown in black; authors' reply is shown in blue.

REVIEWER COMMENTS

Reviewer #1 (Remarks to the Author):

This paper seeks to define the presence and, if present, magnitude of trends in fire danger indices in the western U.S. by elevation. It presents evidence that this may be the case, but that evidence is based on poorly described methods and has caveats altogether undescribed and thus dismissed.

In particular, the authors rely on a gridded climatic dataset (gridMet) that is itself a blend of other inputs that are required to calculate the various fire danger indices. The gridMet documentation online and in Abatzoglou 2013 indicate that the gridded datasets likely used to produce the derived fire danger indices are a combination of NLDAS-2 and PRISM, so rely in part on base data that estimate the temperature and precipitation (and other derived or, in the case of NLDAS, supposedly physically consistent) differences at high elevations from interpolation rather than observation; dramatically fewer high elevation stations exist, and so validation is comparatively limited. The authors make no effort in the current paper to diagnose whether the elevation dependence inferred to be "real" could be instead an artifact of interpolation schemes, changes in data availability, or the "blending" and downscaling processes acting on differentially assimilated data through time.

I am sympathetic that they are using the best there is for these purposes, but also think reanalysis, rescaling, and blending must be shown (or logically defended) to not produce the results as artifacts. I also get it that as a heavily derived variable, ERC can't really be verified, only computed. But the fact remains the higher elevation responses are not validated here – only calculated and trends described as important.

I think the paper should include some methodological upgrades to: (1) more clearly state why the authors believe the trends computed have no artifacts (temporally inconsistent data assimilation / observational / timing / interpolation) contributing to the trends; (2) state how the trends were calculated (and why linear methods are appropriate – a number of these variables lend themselves to non-standard distributions); and (3) describe and then discuss the limitations of the fuel model(s) chosen for ERC.

Response: We appreciate the reviewer's insightful comments, and their critical but positive evaluation of this manuscript.

R(1) – are trends an artifact of the gridMET dataset?

We appreciate that the reviewer is "*sympathetic that [we] are using the best there is for these purposes*", and we made all possible efforts to address the reviewer's comment. Specifically, we have conducted additional analyses and provided conceptual reasoning on why the demonstrated elevation-

dependent fire danger intensification trends in this study are not an artifact of the chosen dataset. Here we present our reasoning:

- (a) We conducted an analysis of fire danger trends across the elevation gradient using available in situ meteorological observations, which confirmed the elevation-dependent fire danger intensification reported in the paper. It is well recognized that long-term high-quality weather observations are sparse in mountainous regions as there are few monitoring sites at higher elevations (also acknowledged by the reviewer). Nonetheless, we attempted to gather observational data to complement the gridded effort. We used vapor pressure deficit (VPD) as the fire danger metric, since VPD calculation only requires daily maximum and minimum temperatures and relative humidity that are commonly observed in various weather stations. The broader suite of observations needed to calculate ERC results in a more limited sample of stations. It is well known that the response of VPD to temperature increase is exponential (see figure below) which inflates trends in lower elevation stations that have higher temperature baselines compared to cooler higher elevation stations. To address this issue, we used linear regression trends in the z-score of VPD in each station ($z = \frac{x - \text{mean}(x)}{\text{std}(x)}$).

[REDACTED]

Figure 2 of Shamshiri and Ismail, 2013

We acquired weather station data using the Meteostat Python library (<https://dev.meteostat.net/python/>), which provides access to open weather and climate data. Historical observations and statistics are obtained from Meteostat's bulk data interface and consist of data provided by different public interfaces, most of which are governmental (see <https://meteostat.net/en/about>). Among the data sources are national weather services like the US National Oceanic and Atmospheric Administration (NOAA). All stations within the boundaries of mountainous ecoregions of the Western US were downloaded for further analysis, among which, we selected stations with at least ten years of continuous and consistent daily data (daily max and min temperature and hourly relative humidity). We calculated daily VPD, and then

averaged them over May through September to calculate average warm-season VPD. We then calculated z-score VPD for warm season averages. We subsequently calculated the linear regression trend slope of z-score VPD as a function of elevation for ecoregions that had at least 10 stations. Obviously, neither 10 years of data is sufficient for temporal trend calculation nor 10 stations across the elevation gradient is enough for robust elevational trend analysis, but given the scarcity of consistent data providing stations, we proceeded with this selection. Table 1 lists the number of stations that were available in each ecoregion, as well as the number of stations with at least 10, 20, and 30 years of consistent observations of daily max/min temperatures and relative humidity.

Table 1. Number of weather stations in each ecoregion, as well as the number of stations with at least 10, 20, and 30 years of continuous daily temperature and relative humidity data

Ecoregion	Number of stations	#Stations with >10 yrs of data	#Stations with >20 yrs of data	#Stations with >30 yrs of data
4: Cascades	1	0	0	0
5: Sierra Nevada	7	7	2	1
11: Blue Mountains	9	6	4	4
13: Central Basin and Range	34	24	16	14
15: Northern Rockies	8	7	2	1
16: Idaho Batholith	3	3	1	0
17: Middle Rockies	16	13	7	6
19: Wasatch and Uinta Mountains	3	2	1	1
20: Colorado Plateaus	16	13	1	1
21: Southern Rockies	17	14	1	1
22: Arizona/New Mexico Plateau	15	15	8	8
23: Arizona/New Mexico Mountains	7	7	1	0
78: Klamath Mountains/California High North Coast Range	5	3	3	3

Only five ecoregions met these loose criteria [10+ stations with at least 10 years of consistent observation], among which four confirmed elevation-dependent fire danger intensification (red shaded areas in the figure below). Please note that none of these elevational trends (positive or negative) reached statistical significance, acknowledging that statistical significance on 10 data points does not mean much!

Figure 1. Slope of z-score of average warm-season VPD temporal trends across elevation, where positive values (red shades) indicate larger intensification of fire danger at higher elevations.

Finally, we provide further details about the stations used in Figure 1 in Table 2 (at the end of the response to reviewer 1 section). These stations are mainly located at airports.

- (b) Elevation-dependent warming has been widely documented by international efforts based on extensive ground observation data (for example see EDW 2015 and the references therein; also see Table 2 below), including in the mountains of the Western US. In most cases, observational studies reported increased rates of warming with elevation.

Table 2. Elevation-dependent warming based on observation data.

[REDACTED]

Source: Pepin et al. 2015

Elevation-dependent warming on the backbone of widespread drying in the fire season in the western US (Holden et al. 2018) is expected to induce elevation-dependent fire danger intensification.

- (c) In a former paper (Alizadeh et al. 2021, PNAS), we demonstrated that normalized burned area from 1984-2019 increased at a higher rate above the 2,500 m elevation mark compared to any other elevation in the Western US. We also demonstrated that background warming has weakened the high-elevation flammability barrier and enabled the upslope advance of fires in the mountainous regions of the Western US. These trends are based on the burned areas determined by the Monitoring Trends in Burn Severity program (Landsat-based products), and are not a byproduct of any interpolation or blending of source data. We acknowledge that a variety of factors, including topography, vegetation continuity, density and type, as well as historical and current anthropogenic factors govern fire activity. But, weakening of the high-elevation flammability barrier played a critical role in inducing a higher rate of increase in burned area at high elevations compared to low elevations. This provides a secondary line of evidence that fire danger is increasing at a higher rate at higher elevations.
- (d) Validation of the gridMET gridded surface meteorological data was conducted against an *extensive* network of weather stations including RAWS, AgriMet, AgWeatherNet and USHCN-2. For more information on validation measures see Abatzoglou (2013). Please note that validation of gridded datasets can be conducted against any available data, which provides a lot more viable stations than what we can use in the presented analysis under item (a). Although gridMET does not explicitly check for temporal homogeneity, it provides one of the best tools out there for this study (as acknowledged by the reviewer). We considered alternative datasets, but they generally did not have the requisite variables needed for fire danger calculations (e.g., NOAA nclimgrid) or were too coarse for elevational relationships (e.g., ERA-5 reanalysis).

Influential studies in the fire literature, including Jolly et al. 2015, used gridded meteorological datasets, which were developed using similar approaches to those used in gridMET, to demonstrate trends in global fire danger. The density of weather stations – and so much so their accuracy and uncertainty – are widely different in various regions. Figure 2 shows the global distribution of Global Historical Climatology Network daily (GHCNd) stations, which indicates a huge gap in Africa and the Amazon, and a low network density in Asia. Lack of dense station network and assimilation of ground-truth (and associated uncertainties) into the gridded meteorological products are shortcomings that the scientific community has accepted.

[REDACTED]

Figure 2. Density of Global Historical Climatology Network daily (GHCNd) stations globally.

We added the following text to the revised manuscript to reflect these points:

We recognize that ground observations of meteorological variables are rare in the high elevations, which might induce uncertainty in the reported trends that are based on a gridded product (gridMET⁵⁰). Our analysis of fire danger trends across the elevation gradient using ground observations, however, confirms the reported findings (Fig. S53) although for a limited number of ecoregions (five) and stations (a total of 79) constrained by data availability. We also note that elevation-dependent warming has been widely demonstrated across the globe, including in the mountains of the western US (Pepin et al. 2015; 2022, and references therein); and this trend on the backbone of widespread drying in the fire season (Holden et al. 2018) is expected to induce elevation-dependent fire danger intensification. Furthermore, Alizadeh et al. 2021 showed that normalized burned area in the high-elevation forests of the western US increased at a higher rate than its low-elevation counterpart from 1984-2019, pointing to a weakened flammability barrier in the high elevations, providing secondary evidence for the herein reported trends.

R(2) – trend calculations

For all variables, we used average daily values in the warm-season (May-September) and estimated least-square linear regression trends over 1979-2020. Obviously, some of these variables may not lend themselves to a normal distribution – although in this case a linear fit seems to be logical; see the Figure 3 below – but the choice of a linear trend supports a straightforward communication of the results. We have added a detailed plot that presents the average warm-season Energy Release Component (ERC) for each year and the linear fit as a supplementary figure (see below, Figure 3). We also included the warm season average for each year in each elevation band for each fire danger index/variable as SI Data. Furthermore, we added text to the revised manuscript to inform the readers that the linear trend choice

is mainly due to simplicity of presentation, and interested audience can use the original data for any other fit.

Figure 3. Annual warm-season average ERC in each elevation band (columns) in each ecoregion (rows), and associated temporal trends.

R(3) – ERC model

We used the Fuel Model G from the US National Fire Danger Rating System 77 (Cohen and Deeming 1985). This model is based on a short needle pine forest with heavy dead fuel. A description of the model is provided below. We note that this choice is to purely reflect climate driven trends, and vegetation impacts (type and availability) on the fire danger is not intended in this study. We selected the fuel model G as a standard, common model across the Western US to ensure comparability of trends within and across ecoregions. We agree that the numerical values of ERC in terms of how they translate into measurable fire behavior (heat release) will vary substantially as a function of fuel, but that is outside of the core questions asked herein. We have added this information to the revised text.

Fuel Model G – Fuel Model G is used for dense conifer stands where there is a heavy accumulation of litter and down woody material. Such stands are typically over mature and may also be suffering insect, disease, and wind or ice damage—natural events that create a very heavy buildup of dead material on the forest floor. The duff and litter are deep and much of the woody material is more than three inches in diameter. The undergrowth is variable, but shrubs are usually restricted to openings. Types to be represented by Fuel Model G are hemlock-Sitka spruce, coastal Douglas fir, and wind thrown or bug-killed stands of lodgepole pine and spruce. (Cohen and Deeming 1985)

References:

- Alizadeh, M.R., Abatzoglou, J.T., Luce, C.H., Adamowski, J.F., Farid, A. and Sadegh, M., 2021. Warming enabled upslope advance in western US forest fires. *Proceedings of the National Academy of Sciences*, 118(22), p.e2009717118.
- Abatzoglou, J.T., 2013. Development of gridded surface meteorological data for ecological applications and modelling. *International Journal of Climatology*, 33(1), pp.121-131.
- EDW, 2015. Elevation-dependent warming in mountain regions of the world. *Nature Clim Change* 5, 424–430. <https://doi.org/10.1038/nclimate2563>
- Holden, Z.A., Swanson, A., Luce, C.H., Jolly, W.M., Maneta, M., Oyler, J.W., Warren, D.A., Parsons, R. and Affleck, D., 2018. Decreasing fire season precipitation increased recent western US forest wildfire activity. *Proceedings of the National Academy of Sciences*, 115(36), pp.E8349-E8357.
- Shamshiri, R. and Ismail, W.I.W., 2013. A review of greenhouse climate control and automation systems in tropical regions. *J. Agric. Sci. Appl*, 2(3), pp.176-183.
- Cohen, J.D. and Deeming, J.E., 1985. The national fire-danger rating system: basic equations. Gen. Tech. Rep. PSW-GTR-82. Berkeley, CA: U.S. Department of Agriculture, Forest Service, Pacific Southwest Forest and Range Experiment Station. 16 p. <https://doi.org/10.2737/PSW-GTR-82>
- Jolly, W.M., Cochrane, M.A., Freeborn, P.H., Holden, Z.A., Brown, T.J., Williamson, G.J. and Bowman, D.M., 2015. Climate-induced variations in global wildfire danger from 1979 to 2013. *Nature communications*, 6(1), pp.1-11.
- Pepin, N., Bradley, R.S., Diaz, H.F., Baraër, M., Caceres, E.B., Forsythe, N., Fowler, H., Greenwood, 318 G., Hashmi, M.Z., Liu, X.D. and Miller, J.R., 2015. Elevation-dependent warming in mountain 319 regions of the world. *Nature climate change*, 5(5), pp.424-430.

Moreover, the paper focuses on ERC almost totally at the expense of other drives of “fire danger”, and as such an important caveat of the work is that it addresses primarily the fire intensity aspects of fire danger, which is much different than assessing the other aspects of fire danger people typically think of

– fire spread, fire weather, etc. – and to which fire managers respond. Stating this in the text would be valuable.

Response: We appreciate this comment. Yes, ERC mainly captures build-up dryness measures of dead and live fuel, and hence potential fire intensity (energy release). The short-duration weather factors responsible for peaks in metrics like the burning index or spread component are of importance to fire operations and management, but are not something we examine in great detail. Please note that we have presented results for burning index (BI; that includes fire spread), 100-hr & 1000-hr dead fuel moisture, vapor pressure deficit, relative and specific humidity, temperature, and reference evapotranspiration in the Supplementary Information. We have highlighted this information in the revised text.

Line by line comments:

Line 36: In-text citation here, but Vancouver style citations elsewhere?

Response: Done. Thanks.

Lines 43-44: “. In particular, more rapid warming of surface air temperature has been documented at higher elevations compared to that of lower elevations in many regions globally^{5,9}” The way this sentence is written implies analysis of observations, but citation 9 is a modeling paper only. I suggest expanding this into two supportable statements, one that it seems to be actually occurring based on data (5) and that the mechanisms for it largely point to snow feedbacks but manifest differently with different LSMs (9). Or similar.

Response: We appreciate this comment and provided another observation-based reference of elevation dependent warming.

Reference:

Diaz, H.F. and Bradley, R.S., 1997. Temperature variations during the last century at high elevation sites. In *Climatic change at high elevation sites* (pp. 21-47). Springer, Dordrecht.

Line 47: There is no single elevation gradient at play here, nor are these changes universally documented, so it should be elevation gradients (plural).

Response: Done. Thanks.

Line 50: NFDRS describes variables that are themselves related to water and energy balance, but a key limitation of NFDRS is that it relies on a few variables, correlated with fire danger as measured by people and parts of energy and water balances, but NFDRS does NOT include the full energy and water balance as they relate to fuels and fire. As such, this is imprecisely stated and needs to be refined to be true. One outcome of changes in energy and water balances is changes in fire danger, but they are not as directly related as this text makes it seem.

Response: We appreciate this comment and modified the text to reflect this point. Revised text reads: *Here we use the fire danger representation in the US National Fire Danger Rating System (NFDRS) (Cohen and Deeming 1985) to investigate trends in fire danger across elevations. We note that NFDRS fire*

danger indices were empirically derived based on mathematical models of fire behavior and they do not fully capture the energy and water balance related to fuels and fire (Cohen and Deeming 1985).

Line 63: Does...trends Do...trends

Response: Done. Thanks.

Line 64: gridMET citation required here – is in methods too, but should be stated here.

Response: Done. Thanks.

Line 67, and in methods. ERC calculations depend on the fuel model chosen. Ecoregions should have different mean vegetation assumptions, and so fuel models could be presumed to vary. In addition, ERC weights larger fuel classes (100hr and 1000hr) more heavily than fine fuels that drive fire spread, and the fire danger aspect of ERC is primarily related to fire intensity (energy released at the flame front). ERC, if evaluated for a universal fuel class (I presume 7G or H, which are common?) across these regions would differentially bias trends of actual fire danger because absolute changes in mean or extreme conditions required to exceed historical thresholds would be quite different for fuel models such as 1 (grass) or 4 (chapparal). It is also true that trends in fuel structure over the period of the study could bias the results as well. It is possible that because the trends are compared for spatially aggregated units and fuels changes were ignored the presence or absence of a trend would remain even if the intercept were incorrect. The authors need to at minimum (1) describe in the methods how they handled the different fuel model possibilities and (2) state why (if it does not) that has no repercussions for results. If it does have implications for results, the authors need to correct the analysis, presumably by specifying different mean fuel models for the ecoregions.

Response: Indeed, ERC calculations are dependent on fuel models if the goal is to directly provide a proxy for potential heat release from a fire. Here, we present a fuel agnostic measure across western landscapes through a single fuel model (model G). The idea is to not conflate heterogeneity in vegetation distribution with heterogeneity in climate trends. Further, while there is some emphasis on ERC in the paper, we show that the general conclusions hold (often more strongly) for other flavors of fire danger metrics that are fuel agnostic. We revised the text to reflect these points.

Line 69-71: ERC has a 7 day memory as typically computed. Especially for coarser fuels (1000hr) this means

Response: We revised this paragraph. Thank you!

Line 154: alfaalfa alfalfa

Response: Done. Thanks.

Line 159-164: Except for the fact that BI includes both, the authors seem to ignore the fact that NFDRS has both ERC and SC – the spread component is part of what determines how likely fires are to grow quickly and burn large areas. ERC will be correlated with the initial potential for intensity to achieve values sufficient to essentially guarantee re=heating and combustion of other fuels in close proximity, but the rate of spread and eventual size of fire is dependent on factors (wind, especially) associated with fire behavior. Critical fire danger days are NOT merely ERC >60, in part because even if all high danger

events $ERC > 60$, not all $ERC > 60$ result in high danger. By focusing on the strongest evident trend, the potential utility of the science is undermined.

Response: In response to this comment and comment 1 of Reviewer 2, we conducted additional analyses using a percentile-based threshold. In our approach, we considered the 75th and 95th percentiles of the long-term daily record in each grid as the threshold for fire danger, estimated high (75th percentile) and extreme (95th percentile) fire danger days for each grid, and averaged them for each elevation band. The results confirmed the original finding of synchronization of fire danger across elevation bands. Please refer to our response to comment 1 of Reviewer 2 for a detailed description of the approach and results.

We also note that several large forest fires, especially in the intermountain areas, have started and grew in the absence of a very high Spread Component (SC). Furthermore, we are not focused on large fire “growth” in this paper, although we agree with the reviewer that the fire weather components that include wind and indices such as SC may more faithfully capture wind driven events. Given our focus on mountain systems, large fire growth days can also occur in the absence of significant wind events given the combination of ample, dry fuels (herein $ERC \geq 60$), and topography.

We have revised the text to reflect these points.

PS: We also note that the choice of ERC was not to provide the strongest evident trend, rather it was due to its wide use in the scientific literature and among the practitioners that we know. In fact, the manuscript reads: *“Warm-season ERC trends were most pronounced at higher elevations and least pronounced at lower elevations in 13 of the 15 studied ecoregions (Fig. 1A). **This trend is even more pronounced for dead fuel moisture (FM100 and FM1000), with all ecoregions showing more marked drying trends at higher elevations (Figs. S3-S4).**”*

Line 211, again 245: “herein revealed”....that language sounds more like marketing than science. I would steer clear of it, especially since the details may, or may not, support revelation.

Response: All such instances are removed from the revised text. Thank you.

Line 263: “We used geospatial analysis...” Is very, very general. Do you mean to say you took daily ecoregional averages by these elevation bands?

Response: We modified the text to: *“We calculated the warm-season average of meteorological and the US National Fire Danger Rating System indices, herein referred to as fire danger indices, using daily values in each grid, and then averaged them for each 500 m elevation band in 15 mountainous ecoregions of the western US from the gridMET⁵⁰ dataset (~4 km resolution).”*

Line 266: “estimating linear trends in warm-season averages”. There are a lot of different ways to estimate linear trends, both magnitude and significance. The choice matters, though, because the nature of the variable responses is by no means guaranteed to be linear and the statistical significance (variability versus magnitude) of the trend is critical. Some specifics here would help make this more presentable.

Response: Please refer to our response to the first comment, under section “R(2) – trend calculations”. Thank you.

Table 2. Description of the stations that were presented in Figure 1.

Ecoregion	Station_id	Lat	Lon	Elevation	Station Description
13	70580	40.8249	-115.792	1567	Elko / Coin
13	72475	38.4266	-113.013	1536	Milford / Yellow Banks
13	72480	37.3667	-118.367	1256	Bishop Airport
13	72485	38.0602	-117.087	1655	Tonopah / Valley View
13	72486	39.3	-114.85	1907	Ely Airport (Yelland Field)
13	72488	39.4833	-119.767	1345	Reno/Tahoe International Airport
13	72572	40.7833	-111.967	1288	Salt Lake City International Airport
13	72575	41.2	-112.017	1362	Ogden-Hinckley Airport
13	72580	40.0664	-118.565	1191	Lovelock / Granite Point
13	72581	40.7333	-114.033	1291	Wendover / Air Force Auxillary Field
13	72583	40.9	-117.8	1312	Winnemucca Municipal Airport
13	72584	40.3757	-120.573	1265	Susanville / Johnstonville
13	74003	40.1667	-112.933	1326	Dugway Proving Grounds
13	KBAM0	40.599	-116.874	1381	Battle Mountain / Rosny
13	KCDC0	37.701	-113.099	1714	Cedar City
13	KCXPO	39.1923	-119.733	1434	Carson / New Empire
13	KHTHO	38.5444	-118.634	1289	Hawthorne / Babbitt
13	KLGU0	41.7913	-111.852	1359	Logan / Greenville
13	KMLDO	42.1704	-112.289	1373	Malad City
13	KNFLO	39.4178	-118.699	1199	Fallon / Fallon Station
13	KP680	39.5167	-115.967	1993	Eureka
13	KPVU0	40.2192	-111.723	1371	Provo / Lakeview
13	KRTSO	39.6682	-119.876	1539	Reno / Reno-Stead

13	KU240	39.3333	-112.583	1414	Delta
17	72577	43.6073	-110.738	1966	Jackson Hole / Moose
17	72686	45.1238	-113.881	1232	Salmon / Baker
17	72770	45.2554	-112.552	1599	Dillon / Bishop Place
17	72772	46.6	-111.967	1180	Helena Regional Airport
17	72773	46.9167	-114.1	976	Missoula International Airport
17	72774	45.9548	-112.498	1692	Butte / Valley Vista Mobile Home Community
17	KBZNO	45.7776	-111.152	1363	Bozeman / Belgrade Village Mobile Home Park
17	KCUTO	43.7333	-103.618	1708	Custer
17	KFWZO	42.5167	-108.783	2588	South Pass / Atlantic City
17	KLLJO	44.5236	-114.218	1546	Challis / Garden City (Historical)
17	KP600	44.55	-110.417	2368	Yellowstone / Lake
17	KSPFO	44.4811	-103.786	1199	Spearfish
17	KWYSO	44.6884	-111.118	2027	West Yellowstone
20	72371	36.9333	-111.45	1314	Page Municipal Airport
20	72470	39.6167	-110.75	1814	Price Carbon County Airport
20	72476	39.1333	-108.533	1481	Grand Junction Walker Field
20	KAIBO	38.2388	-108.563	1811	Nucla / Naturita
20	KAJZO	38.7864	-108.064	1583	Delta / North Delta
20	KBDGO	37.5833	-109.483	1789	Blanding
20	KCNYO	38.755	-109.755	1389	Moab / Crescent Junction
20	KEEOO	40.0488	-107.886	1959	Meeker
20	KHVEO	38.418	-110.704	1355	Hanksville
20	KKNBO	37.0111	-112.531	1484	Kanab
20	KMTJO	38.5098	-107.894	1755	Montrose

20	KRILO	39.5263	-107.727	1688	Rifle / Antlers
20	KVELO	40.4409	-109.51	1609	Vernal / Naples
21	K20V0	40.0537	-106.369	2259	Kremmling
21	KAEJ0	38.8142	-106.121	2423	Buena Vista / Johnson Village
21	KANK0	38.5383	-106.049	2293	Salida / Smelertown
21	KASE0	39.2232	-106.869	2389	Aspen / Woody Creek
21	KAXX0	36.422	-105.29	2554	Angel Fire / Agua Fria
21	KCCU0	39.7833	-106.25	3680	Copper Mountain / Slate Creek
21	KCPW0	37.4514	-106.8	3584	Wolf Creek Pass / Lucky Seven Summer Homes
21	KEGE0	39.6428	-106.916	1996	Eagle County / Gypsum
21	KGUC0	38.5339	-106.933	2341	Gunnison
21	KLAM0	35.8798	-106.269	2186	Los Alamos
21	KLXV0	39.2203	-106.317	3028	Leadville / Stringtown
21	KMYPO	38.4833	-106.317	3667	Monarch Pass / Garfield
21	KSBS0	40.5162	-106.866	2098	Steamboat Springs / Steamboat li
21	KTEX0	37.9538	-107.909	2765	Telluride / Lime
21	KVDW0	41.1567	-105.403	2560	Vedauwoo / Sherman
22	72276	35.65	-109.067	2054	Window Rock Airport
22	72365	35.0333	-106.6	1631	Albuquerque International Airport
22	72374	35.0333	-110.717	1505	Winslow Municipal Airport
22	72462	37.4333	-105.867	2297	San Luis Valley Regional
22	K04V0	38.0993	-106.174	2393	Sagauche / Saguache
22	K4SLO	35.95	-107.083	2195	Star Lake Johnson Ranch / Lagunitas
22	KAEG0	35.1452	-106.795	1779	Albuquerque / Taylor Ranch
22	KFMNO	36.7413	-108.23	1678	Farmington
22	KGNT0	35.1673	-107.902	1993	Grants-Milan

22	KGUP0	35.5111	-108.789	1973	Gallup / Allison
22	KPRC0	34.6544	-112.42	1537	Prescott / Tutt
22	KSAF0	35.6171	-106.089	1935	Santa Fe / Cieneguilla
22	KSJN0	34.5186	-109.379	1748	St Johns / Saint Johns
22	KSKX0	36.4582	-105.672	2163	Taos / Los Cordovas

Reviewer #2 (Remarks to the Author):

Widespread warming/drying is evident, but not consistently so, across western US landscapes over recent decades. For example, underlying/related trends (i.e., in surface temperature, precipitation, evapotranspiration, snow cover, greenness) have been found to vary across the elevational gradient. These changes should be reflected in elevation-related trends in fire danger. Geographic specificity with regard to increased/increasing fire danger is important to factor into fire and land management planning and response aimed at mitigating risk to highly valued resources and assets. The research presented here takes a deeper dive into demonstrated increases in fire danger, focusing on the past four decades of change across the elevation gradient in mountainous regions of the western US. Findings include a remarkably consistent signal of increased warm-season fire danger across all metrics analyzed, with greatest changes over the study period in the highest-elevation band (>3000 m). Estimated numbers of days conducive to large-fire activity increased the most at 2500-3000 m (present in nine of the 15 focal ecoregions). In some ecoregions, including the Central Basin and Range, the estimated average annual number of critical fire-weather days is now essentially equal across the elevation gradient. The work is straightforward (using the gridMET dataset of daily meteorological and NFDRS variables, Level III EPA Ecoregions, and 10-m data from the National Elevation Dataset) and thoroughly yet succinctly reported, with ERC as the focal fire danger index but similar changes visible through the lens of each fire-danger metric (included as supplementary material). (The gridMET dataset has been used extensively to explore a variety of connections between wildfires and climate.) The findings, especially that relatively mesic higher-elevation zones in mountainous regions have become fire receptive over longer within-year periods in recent decades, have important ecological and fire-management implications.

Response: We appreciate the reviewer's time and energy invested in comprehensive and positive assessment of this paper, and their in-depth and constructive comments and suggestions. We have addressed all comments and incorporated the suggestions into the revised manuscript to improve its quality.

My primary concerns/suggestions are as follows:

Data and methodology

Lines 164-165, 283: Brown et al. (2004) is cited as the source of the critical fire danger threshold for used for ERC. As the authors are aware, an ERC of 60 does not mean the same in terms of percentile fire danger across the west or even within an ecoregion. An ERC value of 60 may indicate 95th percentile conditions in one cell, whereas in another cell the 95th percentile ERC value may be 89 – but in both cases the 95th percentile value indicates a comparable level of aridity for that microclimate (e.g., Riley et al. 2013). Figure 2 in Freeborn et al. (2019) shows the wide range of median and maximum ERC(G) values calculated from 39 years of gridded data. This is why contemporary geospatial modeling applications and Fire Danger Operating Plans, for example, are not tied to an absolute index value like ERC = 60 for any of their thresholding, but instead use historical analyses of fire activity and weather to set ERC breakpoints and associated adjective ratings (e.g., “low, medium, high, extreme” fire danger). The authors explain that their “constant threshold, as opposed to a percentile-based approach, is selected herein to warrant consistency and inter-comparability across ecoregions.” But I would like to see a bit more discussion about the pros and cons of the specific ERC = 60 threshold use. Has there been any further investigation since the Brown et al. (2004) publication that supports widespread use of ERC ≥ 60 as a critical fire danger level? For example, have there been efforts that this work could capitalize

on to associate historical large fire activity more precisely with ERC thresholds better tailored to the analysis units herein? Or could the methods here be expanded to make refinements to the ERC threshold as warranted? Perhaps some sensitivity analysis could be performed? Since I expect the media and others will pick up on the reported finding that “the greatest increase in the number of days conducive to large fires occurred at 2,500-3,000 m, adding 63 critical fire days in 42 years”, that seems like a particularly relevant slice of the analysis to look a bit deeper into the relationship between historical fire activity and $ERC \geq 60$.

(Riley, Karin L.; Abatzoglou, John T.; Grenfell, Isaac C.; Klene, Anna E.; Heinsch, Faith Ann. 2013. The relationship of large fire occurrence with drought and fire danger indices in the western USA, 1984-2008: The role of temporal scale. *International Journal of Wildland Fire*. 22: 894-909.)

Response: We appreciate this comment and conducted additional analyses using a percentile-based threshold for all fire danger indices. Percentile-based results confirmed the original reported findings that critical fire danger days increased at a higher rate at higher elevations, which in turn resulted in the synchronization of fire danger across elevation.

We are not aware of newer papers that propose an $ERC \geq 60$ as a threshold for critical fire danger. We, however, note that $ERC \geq 60$ was confirmed in Riley et al. 2013 as a threshold for large fire occurrence. Their Figure 5a (see below) shows empirical cumulative distribution functions (ECDF) of indices, for all conditions [semi-linear] and those associated with large fires [sigmoid-like curve]. As shown here, ERC ECDF has a breaking point at around $ERC = 60$, associated with more frequent large fires.

[REDACTED]

Figure 4. Empirical cumulative distribution functions (ECDF) of ERC, for all conditions and those associated with large fires.

To further examine whether or not $ERC = 60$ is a proper threshold for critical fire danger conditions, we plotted a histogram of ERC at the detection date of fires in the FPA_FOD dataset from 1992-2020 in the 15 mountainous ecoregions of this study:

Figure 5. Distribution of ERC at the detection date of fires. (A) All fires, and (B), (C) fires with a size of >1,000 and >10,000 acres (~400 and 4,000 ha), respectively. ERC = 60 is marked with a red vertical line.

Note that these histograms only show the number of fires at each ERC level, without recourse to the underlying distribution of ERC – e.g., not considering the lower number of $\text{ERC} \geq 60$ days – and hence is not a representation of the probability of fire growth or size as a function of ERC. However, these distributions clearly show that a majority of large ($\geq 1,000$ acres) and very large ($\geq 10,000$ acres) fires had an $\text{ERC} \geq 60$ on their detection date. In fact, 77% and 83% of large and very large fires were associated with $\text{ERC} \geq 60$ in the 15 mountainous ecoregions of Western US. This also holds for the entire Western US (11 most western states in CONUS) with 77% and 85% of large and very large fires being associated with $\text{ERC} \geq 60$ on their detection date. We also note that fire growth may occur on a day different from the detection date, but this is an analysis of 752,496 fires, which lessens the concerns about ERC being or not being highest on the detection date of individual fires.

We also note that – given the background warming and increasing trends in fire danger indices – selecting a percentile-based threshold may result in more pronounced temporal trends compared to a constant threshold. To show this conceptually, please see the following synthetic analysis:

Synthetic data: Generate 40 years of data using a normal distribution with mean μ and standard deviation std , with each year comprising 150 points (~5 months of daily data). We used a $\mu = 50$ for the first year, and linearly increased it at a rate of $20/39$ in each subsequent year. This will add 20 units in 40 years to the average value. We kept a constant $\text{std} = 40$ units for all years.

We now consider a constant threshold of 82 (which corresponds to the ~95th percentile of 40 years of data with a constant mean of 50) and a 95th percentile of the synthetic data (with an increasing mean from 50 in year 1 to 70 in year 40), and estimate trends in critical days in each scenario. We repeat this 100 times. For the constant threshold scenario, the number of critical days increased at a rate of 3.8% per year, whereas for the percentile-based scenario, the number of critical days increased at a rate of 4.9% per year. This difference (higher trend for percentile-based threshold compared to the constant threshold) is statistically significant ($p=0.01$).

For these reasons, we keep a constant threshold analysis in the paper – acknowledging its shortcomings – and provide the percentile-based threshold results in the Supplementary Information. Please see below our percentile-based analysis and results.

In the new analyses, we calculated the 75th and the 95th percentiles of daily ERC (and other variables) values for each grid (from pooled calendar year data for 1979-2020), and used it as the threshold beyond which high and extreme fire danger would occur. We then estimated the number of high and extreme fire danger days in each grid in each year, and averaged the grids in each elevation band to report high and extreme fire danger days in each elevation band. We then used these values for trend analysis. The following figures confirm the originally reported trends with a constant threshold.

Figure 6. ERC threshold associated with the 75th and 95th percentiles, respectively, of daily ERC values from 1979-2020.

Figure 7. Elevation-dependent increase in high fire danger days. (A) Temporal trends in high fire danger days – associated with daily ERC larger than the 75th percentile of long-term record from 1979-2020 – in

each elevation band and ecoregion. **(B)** Slope of temporal trends in high fire danger days across elevation bands. Hatched areas indicate statistically significant trends at the 95% confidence level.

Figure 8. Annual high fire danger days associated with daily ERC larger than the 75th percentile of long-term record from 1979-2020. Decadal average high fire danger days per year from 1981-1990 (blue) and 2011-2020 (orange).

We provide these results for other variables – and for the 95th percentile threshold – at the end of this file. These figures are also provided in the Supplementary Information of the revised paper.

References:

- Alizadeh, M.R., Abatzoglou, J.T., Luce, C.H., Adamowski, J.F., Farid, A. and Sadegh, M., 2021. Warming enabled upslope advance in western US forest fires. *Proceedings of the National Academy of Sciences*, 118(22), p.e2009717118.

Lines 264-265: For those who may not be familiar with EPA ecoregions, how were the “15 mountainous ecoregions” selected? Is “mountainous” an EPA ecoregion attribute?

Response: These are level III Omernick ecoregions that are mountainous (other ecoregions are not mountainous). We revised the text to clarify this point.

Line 275: Which NFDRS version and fuel model was used in calculations? Because ERC is the focus of the main analysis/discussion, it in particular merits further explanation as to what it represents and how it was calculated (e.g., following Freeborn et al. 2015, Jolly et al. 2019).

Response: Fuel model G and NFDRS 77 were used in this study. Additional information was added to the revised text.

Suggested improvements

Figures S9-S18 are missing the ecoregion labels in each box.

Response: Figures are revised. Thank you for noting this.

Line 76: Statement refers to “all fire danger indices” but cites only Figs. S1-S8. ERC data are presented in Fig. 1, which should be referenced here as well.

Response: Added. Thank you!

Line 78: All figures (1, S1-S8) are presented with the same two panels (A&B); suggest referencing Fig. 1A and 1B here as examples “(e.g., Fig. 1A).”

Response: Done. Thank you!

Lines 122-125, 194-196; Fig. 2, S9-S18. The placement of each ecoregion’s plot in these figures appears based on (information-free) ecoregion number: from lowest-to-highest, top-to-bottom, starting on the right panel. Have you considered placement that aligns better with the N-S-E-W location (or just latitude) in the western US? For example, Arizona/New Mexico Mountains and other southern ecoregions toward the bottom of the figure and Canadian Rockies, North Cascades, et al. toward the top? There wouldn’t be perfect correspondence, of course, (you could use the current logic for “ties”) but it would help convey general geographic differences in ERC values (e.g., appearing higher in the south vs. north) and how decadal averages have changed in more of a geographic context.

Response: We appreciate this comment. There is in fact a north-south, west-east order in the ecoregion numbering system:

[REDACTED]

Figure 9. Level III Omernick ecoregions of the Western US.

We adopted the ordering based on this system, and have used a similar pattern in previous papers.

Clarity and context

Consider including a map showing the different elevation bands from the NED for the ecoregions used in the analysis. Or at least a map and/or size estimate of the area reported to have had the greatest increase in the number of days conducive to large fires (i.e., 2,500-3,000 m within the nine ecoregions).

Response: We appreciate this comment and included a table in the Supplementary Information that provides details of surface area (km²) encapsulated in each elevation band in each ecoregion.

Table 3: Surface area (km²) encapsulated in each elevation band in each ecoregion

Ecoregion #	0-500	500-1000	1000-1500	1500-2000	2000-2500	2500-3000	>3000
4	7025	17145	19437	13074	1963	143	66
5	320	5867	10536	14094	10632	6842	4812

11	543	12534	38748	16857	2054	174	0
13	0	29	99301	149569	50607	8016	1267
15	1078	33126	34921	12331	503	1	0
16	25	1320	9455	22146	21897	5104	335
17	0	469	23709	48052	53384	30459	8387
19	0	0	190	7210	18518	14644	5130
20	0	15	24004	72597	35980	3952	26
21	0	0	16	6712	48048	54798	36126
22	0	368	9982	90060	45204	1243	1
23	0	1192	16380	39758	44376	8922	278
41	0	122	4344	9620	4617	175	1
77	3135	8140	9931	7295	1843	33	2
78	8961	20517	13460	4759	652	8	0

Line 191: Some results are indicated to reflect “disappearance of topographical relief in a warming climate.” While there may be lessened “relief” from fire danger with elevation gain (cool double meaning), the elevational gradient persists. So perhaps reword to say “disappearance of topographical relief in a warming climate from a fire-danger standpoint” – or something to that effect. Referred to as “fire danger relief” in Line 209.

Response: We modified this sentence to “indicating lessened topographical fire danger relief in a warming climate”. Thank you!

Line 208: Is “synchronization” the best term here? It makes it seem as though things are happening all in unison now. But it’s more nuanced than that. Pretty strong dryness gradients still exist in most ecoregions.

Response: We believe that the results with a constant threshold point to synchronization of the fire danger across the elevation gradient. The gradients are only weakly being changed (see Figure 2 in the main text), but what is changing is the number of days that vegetation is flammable at once across the elevation (see Figure 5 in the main text). A percentile-based analysis, however, would potentially undermine this statement.

References

References 52 and 53 citations of the same manuscript.

Response: Thank you for noting this. Repeated reference was removed.

Lines 162-165: Brown et al. (2004) is cited to support the assertion that the ERC = 60 threshold is “used by fire managers for devising management strategies and planning.” However, this appears based on a statement by Brown et al. (2004) that “conversations with fire specialists independent of this study indicate that ERC values of 40 and 60 might be useful thresholds that can be related to management strategies and planning.” Historical fire-weather analyses have advanced considerably over the last ~20

years. In keeping with previous comments, I would like to see a more contemporary citation that supports the assertion that an ERC = 60 threshold is used in management or planning applications today. Looking at the relevant fire danger plots here, https://www.predictiveservices.nifc.gov/fuels_fire-danger/national_fire_danger.html, could be informative, but they appear to be mostly based on NFDRS 2016/NFDRSv4 and Fuel Model Y, which goes back to the need to understand how ERC was calculated for this analysis.

Response: Please refer to our response to your first comment. We have added percentile-based analyses, and provided additional analyses to support the selection of ERC \geq 60 as a critical fire danger condition. We are not aware of any newer references to support this threshold selection, but our own analysis supports it and percentile-based analyses corroborate findings based on the constant threshold of ERC \geq 60.

High fire danger days based on percentile-based analysis, using the 75th (25th) percentile as the threshold:

Figure 10. Elevation-dependent increase in high fire danger days. (A) Temporal trends in high fire danger days – associated with daily VPD larger than the 75th percentile of long-term record from 1979-2020 – in each elevation band and ecoregion. **(B)** Slope of temporal trends in high fire danger days across elevation bands. Hatched areas indicate statistically significant trends at the 95% confidence level.

Figure 11. Annual high fire danger days associated with daily VPD larger than the 75th percentile of long-term record from 1979-2020. Decadal average high fire danger days per year from 1981-1990 (blue) and 2011-2020 (orange).

Figure 12. Elevation-dependent increase in high fire danger days. (A) Temporal trends in high fire danger days – associated with daily FM100 smaller than the 25th percentile of long-term record from 1979-2020 – in each elevation band and ecoregion. (B) Slope of temporal trends in high fire danger days across elevation bands. Hatched areas indicate statistically significant trends at the 95% confidence level.

Figure 13. Annual high fire danger days associated with daily FM100 smaller than the 25th percentile of long-term record from 1979-2020. Decadal average high fire danger days per year from 1981-1990 (blue) and 2011-2020 (orange).

Figure 14. Elevation-dependent increase in high fire danger days. (A) Temporal trends in high fire danger days – associated with daily FM1000 smaller than the 25th percentile of long-term record from 1979-2020 – in each elevation band and ecoregion. (B) Slope of temporal trends in high fire danger days across elevation bands. Hatched areas indicate statistically significant trends at the 95% confidence level.

Figure 15. Annual high fire danger days associated with daily FM1000 smaller than the 25th percentile of long-term record from 1979-2020. Decadal average high fire danger days per year from 1981-1990 (blue) and 2011-2020 (orange).

Extreme fire danger days based on percentile-based analysis, using the 95th (5th) percentile as the threshold:

Figure 16. Elevation-dependent increase in extreme fire danger days. (A) Temporal trends in extreme fire danger days – associated with daily ERC larger than the 95th percentile of long-term record from 1979-2020 – in each elevation band and ecoregion. (B) Slope of temporal trends in extreme fire danger days across elevation bands. Hatched areas indicate statistically significant trends at the 95% confidence level.

Figure 17. Annual extreme fire danger days associated with daily ERC larger than the 95th percentile of long-term record from 1979-2020. Decadal average extreme fire danger days per year from 1981-1990 (blue) and 2011-2020 (orange).

Figure 18. Elevation-dependent increase in extreme fire danger days. (A) Temporal trends in extreme fire danger days – associated with daily VPD larger than the 95th percentile of long-term record from 1979-2020 – in each elevation band and ecoregion. (B) Slope of temporal trends in extreme fire danger days across elevation bands. Hatched areas indicate statistically significant trends at the 95% confidence level.

Figure 19. Annual extreme fire danger days associated with daily VPD larger than the 95th percentile of long-term record from 1979-2020. Decadal average extreme fire danger days per year from 1981-1990 (blue) and 2011-2020 (orange).

Figure 20. Elevation-dependent increase in extreme fire danger days. (A) Temporal trends in extreme fire danger days – associated with daily FM100 smaller than the 5th percentile of long-term record from 1979-2020 – in each elevation band and ecoregion. (B) Slope of temporal trends in extreme fire danger days across elevation bands. Hatched areas indicate statistically significant trends at the 95% confidence level.

Figure 21. Annual extreme fire danger days associated with daily FM100 smaller than the 5th percentile of long-term record from 1979-2020. Decadal average extreme fire danger days per year from 1981-1990 (blue) and 2011-2020 (orange).

Figure 22. Elevation-dependent increase in extreme fire danger days. (A) Temporal trends in extreme fire danger days – associated with daily FM1000 smaller than the 5th percentile of long-term record from 1979-2020 – in each elevation band and ecoregion. (B) Slope of temporal trends in extreme fire danger days across elevation bands. Hatched areas indicate statistically significant trends at the 95% confidence level.

Figure 23. Annual extreme fire danger days associated with daily FM1000 smaller than the 5th percentile of long-term record from 1979-2020. Decadal average extreme fire danger days per year from 1981-1990 (blue) and 2011-2020 (orange).

REVIEWERS' COMMENTS

Reviewer #1 (Remarks to the Author):

I am satisfied with the authors' responses to my review comments on the prior version of the manuscript. I think this will make a good contribution to the literature on climatically driven changes in fire activity in the western U.S.

Reviewer #2 (Remarks to the Author):

I appreciate the authors' thoughtful responses to the review comments. I am satisfied with the additional considerations they have taken and feel that this is strong work.

Point-by-Point Response to Review Comments

The authors would like to thank the two anonymous reviewers for supporting the publication of this manuscript in Nature Communications. Reviewer's comments are shown in black; authors' reply is shown in blue.

REVIEWERS' COMMENTS

Reviewer #1 (Remarks to the Author):

I am satisfied with the authors' responses to my review comments on the prior version of the manuscript. I think this will make a good contribution to the literature on climatically driven changes in fire activity in the western U.S.

Thank you for your positive evaluation of this manuscript and your support of its publication in Nature Communications.

Reviewer #2 (Remarks to the Author):

I appreciate the authors' thoughtful responses to the review comments. I am satisfied with the additional considerations they have taken and feel that this is strong work.

Thank you for positive evaluation of this manuscript and your support of its publication in Nature Communications.